# Influence of the Enzymatic Hydrolysis Using Flavourzyme Enzyme on Functional, Secondary Structure, and Antioxidant Characteristics of Protein Hydrolysates Produced from Bighead Carp (*Hypophthalmichthys nobilis*)

**DOI:** 10.3390/molecules28020519

**Published:** 2023-01-05

**Authors:** Kamal Alahmad, Anwar Noman, Wenshui Xia, Qixing Jiang, Yanshun Xu

**Affiliations:** 1State Key Laboratory of Food Science and Technology, School of Food Science and Technology, Collaborative Innovation Center of Food Safety and Quality Control in Jiangsu Province, Jiangnan University, Wuxi 214122, China; 2Department of Food Science and Technology, Faculty of Agriculture, University of Alfurat, Deir Ezzor, Syria; 3Department of Agricultural Engineering, Faculty of Agriculture, Foods and Environment, Sana’a University, Sana’a 13060, Yemen

**Keywords:** flavourzyme enzyme, bighead carp, enzymatic hydrolysis, CD secondary structure, FTIR, functional characteristics, antioxidants activities

## Abstract

In the current study, bighead carp fish were used in conjunction with the flavourzyme enzyme to obtain (FPH) fish protein hydrolysates. The optimum conditions of the hydrolysis process included an enzyme/substrate ratio of 4% and a temperature of 50 °C and pH of 6.5. The hydrolysis time was studied and investigated at 1, 3, and 6 h, and the (DH) degree of hydrolysis was recorded at 16.56%, 22.23%, and 25.48%, respectively. The greatest yield value was 17.83% at DH 25.48%. By increasing the DH up to 25.48%, the crude protein and total amino acid composition of the hydrolysate were 88.19% and 86.03%, respectively. Moreover, more peptides with low molecular weight were formed during hydrolysis, which could enhance the functional properties of FPH, particularly the solubility property ranging from 85% to 97%. FTIR analysis revealed that enzymatic hydrolysis impacted the protein’s secondary structure, as indicated by a remarkable wavelength of amide bands. Additionally, antioxidant activities were investigated and showed high activity of DDPH radical scavenging, and hydroxyl radical scavenging demonstrated remarkable activity. The current findings demonstrate that the functional, structural, and antioxidant characteristics of FPH might make it an excellent source of protein and suggest potential applications in the food industry.

## 1. Introduction

Carp were found and reported early in different parts of China and have spread to various areas, including North American areas such as Ohio and the Tippecanoe River. The global harvest of carp was over 3300 kilotons in 2018 [1]. In general, carp appear silver on the back and sides, and the bellies are grey or creamy. Carp is a popular and affordable ingredient in East Asia. Thus, the quality of bighead carp is a priority concern for producers and consumers regarding fish planting, harvesting, and marketing [2].

Environmentally, carp could clean and filter lakes and freshwater territories using their rakers and could feed on a wide variety of different organisms such as algae, insects, and some freshwater plants. However, carp have a high tolerance in challenging conditions, and they can gather in groups to overcome environmental stressors [3]. Moreover, the size of bighead carp (between 7 and 15 pounds) is preferable for consumption, but the large or small size might affect the cost in the fish market. Generally, fish protein is remarkably digestible, with sufficient amounts of crucial amino acids and peptides that are lacking in various other food sources. In other words, freshwater fish proteins contain remarkable values of antioxidants and antimicrobials due to sufficient quantities of amino acids [4].

Carp consumption is highly recommended nutritionally due to its chemical composition of diverse types of amino acids (including essential and non-essential amino acids), abundant fatty acids (including docosahexaenoic acid and eicosapentaenoic acid (DHA)), and abundant vitamins (including vitamin D3, E, and vitamin (B) complex including B2, B6, and B12) [5]. Obviously, there is a major need to use fish resources with more focus and attention on ingredients, including peptides, amino acids, and some important fatty acids in seafood resources [6]. Various techniques could be used to produce protein hydrolysates, such as ultrasonic sonication, microwave-assisted techniques, and using organic or chemical solvents.

The conventional pathway to obtain protein hydrolysates uses concentrated acid or base to separate large peptide molecules at the optimum temperature and pressure to derive small peptides and the desired amino acids [7]. The enzymatic hydrolysis process targeting seafood protein had remarkable potential to be applied in different food industries compared with other hydrolysis methods. Enzymatic hydrolysis might provide prospective nutritional and functional products for diverse uses in food applications [8]. The current hydrolysis process could enhance and release various desirable properties of freshwater fish protein. Additionally, small active peptides that are capable of being efficiently used as antioxidants, antimicrobials, and anti-cancer agents are released by enzymatic hydrolysis, which can improve and regulate various bodily functions and the immune system [4]. Various areas of food, pharmacy, and cosmetics use active fractions of hydrolyzed peptides in their industries [8].

The modification of the enzymatic process of proteins utilizing selected protease enzymes to cleave large peptides into small fractions has been extensively applied in the food industry. Many researchers have investigated hydrolysates’ functional and morphological properties over the past 15 years [9]. Exo- and endo-proteases were applied to obtain FPH, including Ficin, Papain, bromelain, trypsin, alcalase, protamex, validase, neutrase, and flavourzyme [7]. The advantage of protease enzymes is that they can be produced from different sources such as microorganisms, plants, and animals. Protease enzymes are used in synthesis, catalysis, and in different applications of food biotechnology to produce active peptides, provide essential amino acids, and improve the solubility and digestibility of food ingredients [10].

Flavourzyme is a fungal enzyme commercially produced by Aspergillus oryzae, with remarkable properties in terms of generating amino acid sequences and active peptide fractions, and it has a clear influence on some bioactive compounds such as antioxidants and antimicrobials. Flavourzyme is a mixture of exo- and endo-peptidase, which cleaves the peptide bonds within and at the ends of protein molecules [11].

Use of protease enzymes confers better hydrolysis conditions than other methods in terms of achieving and producing high-quality products of FPH. Thus, hydrolysis by enzymes might be an appropriate and applicable technique to enhance the characteristics of FPH and maintain nutritional properties in food applications [7]. The type of substrate, pH of the reaction, temperature, concentration of enzymes, and reaction time are the most important factors influencing the final product of FPH. Furthermore, the optimum conditions of the mentioned factors can smoothen the enzymatic process and lead to high-quality production of FPH [12].

However, additional factors play important roles in terms of FPH production such as the specificity of the protease enzyme, molecular weight, degree of hydrolysis, protein source, and peptide sequences, and the technique applied to obtain the protein hydrolysates could also influence the nutritional and biological value of the final FPH [13]. To enhance the functional characteristics and improve the solubility of protein in different seafood sources, the process of enzymatic hydrolysis could be applied to obtain a desirable FPH. Emulsification properties, foaming capacity, oil and water holding capacity, anti-cancer, antimicrobial, and antioxidant values could be enhanced and increased through the hydrolysis process [14]. Using protein hydrolysates as a nutrition source has spread widely in different food fields in the last two decades, and it might be applied as a medicine for cancer or bacterial diseases and enhance immunity against various gastrointestinal effects [8]. FPH possess different bioactive compounds including antioxidants, which can be applied against different sicknesses or used as food additives to improve the immune system; thus, different studies indicated that the existence of antioxidants in FPH products might increase defense against heart and cancer diseases [15].

The production of FPH with high activity of antioxidants plays a remarkable role in improving bodily functions, and it is applied in various areas of the food industry. Thus study of these properties in FPH could provide an overall comprehensive understanding of functional properties and food additives [16]. However, the current work has studied and investigated the optimum factors, e.g., (E/S)% ratio, pH, temperature, and time of incubation of flavourzyme enzyme, to produce FPH from bighead carp (*Hypophthalmichthys nobilis*) cultivated from the Yangtze River in Jiangsu province, which has not been reported before. We then analyzed some structural, functional, and antioxidant properties of the final product for potential application in food industries as additives or supplements.

## 2. Results and Discussion

### 2.1. Optimization of Enzymatic Hydrolysis Factors

The influence of optimization factors of enzymatic hydrolysis on the degree of hydrolysis using the flavourzyme enzyme is demonstrated in Figure 1. The impact of enzyme/substrate ratio on the DH was investigated at various levels from 1% to 5%, and the results are displayed in Figure 1A. At an enzyme concentration of 1%, the DH value was 15.84%, and by increasing the concentration to 2% and 3%, the DH was 19.6% and 20.83%, respectively.

The highest DH value was 22.6% at 4%, which was the optimum E/S ratio in this study. However, no significant variation was noted when the enzyme/substrate ratio was increased to 5%, which may cause enzyme aggregation, inhibit protein substrate diffusion, and finally decrease the enzymatic hydrolysis process because of the lack of catalytic sites of the targeted enzyme [8]. Serajul Islam and others found that the optimum E/S% for the flavourzyme enzyme to obtain protein hydrolysates from a Chinese pond turtle substrate was 4% [17]. Moreover, the effect of pH on the DH was studied and displayed in Figure 1B. The current study showed that the optimum degree of hydrolysis was 21.8%, achieved at pH 6.5, while lower DH values were observed at other pH levels. These results are similar to those of Noman, who found that the optimum pH value using flavourzyme enzyme was 6 to 7 in hybrid sturgeon muscles [18]. The optimum pH might change according to the substrate and E/S ratio applied in hydrolysis. Thus, changing pH values may cause denaturation in protein substrates and then reduce the capacity of the substrates to react and combine with active sites of the targeted protease [18].

During the enzymatic hydrolysis process, reaction temperature plays a prominent role through its impact on the protein substrate and enzyme activity during the hydrolysis. The influence of five temperature measures from 30 °C to 70 °C was evaluated, and the results are shown in Figure 1C. Moreover, at 50 ℃ the DH value was 22.68%, which was the highest value compared with other temperature levels. Exactly the same optimal temperature value was reported by Noman and others in the work of hybrid sturgeon using flavourzyme enzyme [18]. However, increasing the reaction temperature to 60 °C and 70 °C showed low values of DH, which may be attributed to protein aggregation in the substrate and a decrease in the ability of the protease enzyme to break down large protein into small peptides [19].

Eventually, the degree of hydrolysis was studied and measured seven times from (1 h to 7 h) under the optimum factors studied previously (Figure 1D). In the first hour, the hydrolysis process was remarkably fast, and the DH was 16.56%; the second and third hour continued with high rate of hydrolysis, with DH 19.48% and 22.23%, respectively. Furthermore, the rate of hydrolysis was slow from 3 h to 7 h, with no significant variation of DH between 6 h and 7 h, which were 25.48% and 25.53%, respectively. The hydrolysis process was terminated at 6 h. Ovissipour and others declared that the reduction in hydrolysis rate after 2 to 3 h was due to the reduction of peptide molecules in the substrate, which impacts the hydrolysis process [20]. The degree of hydrolysis at 1, 3, and 6 h, which recorded 16.56%, 22.23%, and 25.48%, respectively, were chosen for further study in this research.

### 2.2. Proximate Analysis of Protein Hydrolysates

The results of the chemical composition analysis of bighead carp protein hydrolysates obtained under optimum conditions using the flavourzyme enzyme are presented in Table 1. According to the relevant results, no significant variation was noted in moisture content for the three hydrolysates, and the result was approximately similar to those reported by Noman for using flavourzyme to obtain hydrolysate from hybrid sturgeon muscles [18]. At DH 25.84%, the protein value was the highest at 88.19%, while the lowest protein content was 83.61% at DH 16.56%.

The protein content variation may be related to different DHs under different hydrolysis process times; thus, the high value of the degree of hydrolysis leads to high protein content in hydrolysate products. The current results showed that various hydrolysates contained high protein content, which might contribute to the elimination of insoluble protein molecules in terms of more soluble protein parts being released during enzymatic hydrolysis, which leads to high solubility in fish protein hydrolysates [21]. Noman and others recorded 85.36% of protein content in the hydrolysate using flavourzyme with a hybrid sturgeon substrate [18], while Alahmad and others found that the protein content ranged between 78% and 85% using ficin enzyme with a bighead carp substrate [7]. However, the protein content in the hydrolysates obtained from Chinese sturgeon fish using alcalse enzyme was between 81% and 86% [22].

Furthermore, fat content was recorded to be 2.98%, 3.34%, and 2.04% for the degree of hydrolysis of 16.56%, 22.23%, and 25.48%, respectively. No significant difference was noticed in fat content at DH 16.56% and 22.23%. However, the DH 25.48% showed a significant variation in fat content compared with lower DHs. Noman and his colleagues found that the fat content in protein hydrolysates obtained from Chinese sturgeon ranged from 7% to 11% [22], and some researchers in 2014 found that fat content was averaged between 4.6% to 20.5% in the protein hydrolysates of Mediterranean fish [23]. As observed in this work, the increase of DH increased the protein content of hydrolysates; conversely, the fat content decreased. Moreover, our study showed that the fat content in hydrolysates achieved at DH 25.48% was lower than that produced at lower DHs. Obviously, there is a contrasting relationship between protein and fat content in hydrolysate products.

Additionally, the hydrolysates with a high amount of lipid might be related to the lipid droplets found in the supernatant after the centrifugation process. Finally, the ash content of hydrolysate products produced from bighead carp using the flavourzyme enzyme is shown in Table 1. The data show that ash content was recorded to be 6.21%, 5.92%, and 4.85% at DH values of 16.56%, 22.23%, and 25.48%, respectively.

The ash content found at DH 16.56% and 22.23% was significantly higher than the ash content in the hydrolysate produced at DH 25.48%. These results were lower than those found by Noman and others for hybrid sturgeon with the flavourzyme enzyme [18] and similar to the results Anwar Noman and his colleagues provided for Chinese fish sturgeon with alcalase enzyme [22]. In general, the ash content in the hydrolysate products is related to the minerals in the substrate as well as the protease enzyme that is applied for the enzymatic process.

### 2.3. Yield Measurement

The yield results in the current study are displayed in Table 1. The yield of FPH was recorded to be 17.83% at the DH 25.48%, while the other values were 17.10% and 13.08% at DH 22.23% and 16.56%, respectively. There was a significant difference between yield values at various DHs. The recorded data were similar to those reported by Noman in Chinese sturgeon fish with papain enzyme [8] and higher than those achieved under optimal conditions of using alcalase enzyme at different DHs [22]. Different yield values may be attributed to enzymatic conditions, including incubation time, and the substrate materials play an important role in achieving various yields during hydrolysis.

### 2.4. Color Determination

Color is potentially the most crucial sensory property in food industries. The color of food products gives consumers a quick impression of the products’ freshness, flavour, and quality. The color values (L*, a*, and b*) are displayed in Table 1. The hydrolysate samples obtained at 1, 3, and 6 h tended to whiteness (L* = 86.89, 84.06, and 83.98), while yellowness was averaged (b* = 15.81, 17.42, and 17.98) for the hydrolysate products at DH 16.56%, 22.23%, and 25.48% respectively. Somewhat similar results were recorded by Alahmad and others in protein hydrolysates of bighead carp fish using the ficin enzyme [7]. Based on the recorded data, no noticeable significant variations were observed in the processed samples at various DHs. Furthermore, in FPH, color values might be attributable to enzymatic hydrolysis conditions, including the reaction time and the type of protease enzyme as well as the content of fish muscles used to obtain the hydrolysate products [24].

### 2.5. Water Activity

Water activity is a function of composition and a function of temperature. The a_w_ in food is related to the boiling and freezing points. Water activity ranges from 0 to 1. The water activity concept refers to food stability in terms of water existing and molecular mobility. The a_w_ plays a vital role in biological processes and chemical reactions in food products. The values of water activity are presented in Table 1. In the current study, the best water activity value was 0.27 at DH 25.48%, while the water activity values at DH 22.23% and 16.56% were 0.34 and 0.36, respectively.

Moreover, these results showed a lower value of a_w_, which might indicate remarkable stability and increase food shelf life. A value of a_w_ between 0.50 to 0.60 could increase the food hardness and give low flavour quality. On the other hand, increasing a_w_ from 0.70 to 0.85 could promote chemical reactions and supply a suitable environment for the growth of bacteria and fungi [25].

### 2.6. Amino Acid Composition

The amino acid profile in fish protein hydrolysates produced using the flavourzyme enzyme under optimized conditions is displayed in Table 1. Under different degrees of hydrolysis, 16.56%, 22.23%, and 25.48%, total amino acids were recorded to be 81.15, 83.05, and 86.03 g/100 g protein, respectively. The total amino acid content values were significantly different (*p* < 0.05). Generally, increasing hydrolysis time leads to an increase in the DH, which impacts the total amino acid content of the hydrolysate samples. The increase might be attributed to the peptide molecules in protein hydrolysates and then boosting the TAA concentration.

A total of 17 amino acids were found in hydrolysate samples, including 10 essential amino acids and 7 non-essential amino acids; leucine, lysine, arginine, aspartic acid, and glutamic acid were found in high amounts compared to other amino acids. These findings were noticeably higher than those reported by Noman in Chinese sturgeon fish using Alcalse enzyme [22]. Total essential amino acids TEAA were 45.34, 46.93, and 49.81 g/100 g protein at DH 16.56%, 22.23%, and 25.48%, respectively. The high amount of essential amino acids in FPH could be used as protein additives to support some meals and food products in food industries [26]. In the current study, the variation in amino acid content under different degrees of hydrolysis compared with those reported in previous studies might be attributed to various impacts such as fish species, fishing season, sex of the fish, fish age, or feeding system [8].

### 2.7. Molecular Mass Distribution

Results of the analysis of the molecular mass profiles of fresh samples compared with FPH obtained from bighead carp at different degrees of hydrolysis using the flavourzyme enzyme in terms of optimum conditions are presented in Figure 2. In comparison with the untreated sample, the hydrolysate samples showed small peptide molecules. This is attributable to the enzymatic hydrolysis process, which significantly breaks down macro peptides during the reaction. Tiny peptides with different molecular masses were investigated and observed in FPH samples obtained at DH 16.56%, 22.23%, and 25.48%. The results of the molecular mass distribution analysis revealed percentages of peptide molecules ˂1000 Da of 88.09%, 94.72%, and 97.87% for the hydrolysate products generated at DH 16.56%, 22.23%, and 25.48%, respectively. During the hydrolysis process, more small peptides were released with small molecular mass as DH and hydrolysis time increased. Furthermore, the results of mass distribution of peptides under 1000 Da in hydrolysates of bighead carp using ficin enzyme at different DH were recorded to range from 82% to 95% [7] and from 66% to 84% in Chinese sturgeon using alcalse [22]. The values of the current study were similar to the findings presented by Noman in FPH produced from Chinese sturgeon using papain enzyme at different hydrolysis times [8]. Some researchers found that utilizing various protease enzymes to produce hydrolysates from salmon fish might lower the molecular weight values from ˃6000 Da to small fractions of peptides (100 to 600 Da) [27].

### 2.8. Scanning Electron Microscopy Evaluation

The SEM results of the analysis of fish protein hydrolysates produced from bighead carp using the flavourzyme enzyme are shown in Figure 3. Regarding the obtained data, after the hydrolysis process, the SEM images revealed that the large molecular peptides were catalyzed into small peptides, and then a reduction in the particle size in (a–c) treated samples of protein hydrolysate products occurred at DH 16.56%, 22.23%, and 25.48% respectively. Furthermore, the SEM images of micrographs of the targeted samples revealed that some particles had rectangular and circular shapes with different angles, which might be related to the influence of the enzymatic process along with water evaporation of hydrolysates during the freeze-drying process [7]. Moreover, the results observed in this research were somewhat similar to those reported by Elavarasan and others, who found that large molecular peptides degrade during the enzymatic hydrolysis process into small-weight peptides and, finally, decrease the size of the targeted samples [28].

### 2.9. Electrophoretic Profile Analysis

The impact of protein enzymatic hydrolysis on the treated samples evaluated using electrophoretic analysis (SDS-PAGE) is displayed in Figure 4. The results revealed that due to the enzymatic hydrolysis process, the peptides with high molecular weight disappeared, implying that using the flavourzyme enzyme had a considerable influence on peptide cleavage in hydrolysate products. Additionally, from SDS-PAGE images, no remarkable differences were noticed between fish protein hydrolysates at DH 16.56%, 22.23%, and 25.48%. Furthermore, most of the peptide bonds in the hydrolysate samples were under 1 kDa with low molecular weight due to the enzymatic impact. However, these results are relatively similar to those reported by [29] in protein hydrolysates of freshwater sailfin catfish *(Pterygoplichthys pardalis)*. Moreover, Fadimu and others found the same results without significant differences in the treated samples using protamex enzyme in the hydrolysis of a lupin protein substrate [13].

### 2.10. FTIR Spectra of Hydrolystes

FTIR spectroscopy is a standard spectroscopic method to determine and evaluate protein hydrolysates and to structurally characterize peptide bonds that are related to structural data existing in spectra of Fourier transform infrared. The aggregations of amino acid sequences of the peptide fractions and protein molecules possess eight to nine unique infrared absorbed bands, defined as the amide bands [30]. The Fourier transform infrared bands are given by amide bands that link and tighten the amino acids. The FTIR absorbance bands of the hydrolysates obtained at various DHs are displayed in Figure 5. From the observed data, the areas of amide absorption of the treated samples were demonstrated at 1521 cm^−1^ (amide I), 1457 cm^−1^ (amide II), and 1392 cm^−1^ (amide III) for the hydrolysate obtained at DH 16.56%, while the absorbance bands of the hydrolysate at DH 22.23% were at 1575 cm^−1^ (amide I), 1455 cm^−1^ (amide II), and 1393 cm^−1^ (amide III). DH 25.48% showed amide bands at 1575 cm^−1^ (amide I), 1457 (amide II), and 1394 cm^−1^ (amide III). According to the results, amide I and II absorption regions for all hydrolysate samples were remarkably different, which might be attributed to the impact of enzymatic hydrolysis factors, including temperature and hydrolysis time, on the protein configuration of hydrolysate products. The current results were similar to Noman’s results [30] in fish sturgeon using two protease enzymes. Amide I and II band areas are highly linked to the protein secondary structure, which can be influenced by enzymatic process factors, particularly the reaction temperature and the protease type used for hydrolysis.

Furthermore, the substrate quality and the type of protease used for hydrolysis to release more fractions of peptides and amino acids would influence the water phase compositions and result in considerable variances in infrared absorbance areas [31]. Based on previous studies, amide I showed at 1550 to 1650 cm^−1^ [32], amide II showed at 11,450 to 1600 cm^−1^ [33], and amide III showed at between 1200 to 1400 cm^−1^ [34]. Generally, amide I is the most important factor to indicate changes in protein secondary structure and influence the various compositions of protein structure such as β-turn, β-sheet, and α-helix. However, amide III bands could be used to detect the protein structure based on its spectra area and give a fairly accurate result in the infrared spectrum. The difference between amide I and amide III is that amide III consists of polypeptides and different components of secondary structure compared with the amide I spectrum absorbance area [35]. The absorption region of amide I impacts the stretching vibration of the carbonyl group C=O chemical bond. In contrast, the absorptions related to amide II lead to extending the bending vibration of N-H bond. Finally, the amide III absorption area leads to stretch N-H and C-N stretching vibrations with the assistance of carbonyl group C=O [36].

### 2.11. Functional Properties of FPH

#### 2.11.1. Protein Hydrolysate Solubility

At different DHs, the solubility of protein hydrolysates under various pH values (ranging from 2 to 10) is displayed in Figure 6. The solubility of hydrolysate products remarkably influences the various functions of the protein and peptide fractions. Using hydrolysate products in the food industry is very important in the applications of foams and emulsifications to boost and improve the functional characteristics of food products [37]. In the current study, the results showed remarkable solubility values of the hydrolysate samples, averaging 84% to 97% at different pH levels, with the highest value recorded to be 97.4% at DH 25.48% and pH 6. The variance was significant (*p* < 0.05). On the other hand, at DH 16.56% and 22.23%, no significant variation was observed in solubility values.

The explanation for high solubility values in protein hydrolysates is related to enzymatic hydrolysis, fractionating large protein molecules into small peptides with low molecular mass, and then increasing the solubility in different hydrolysis products [38]. Protein molecules have an isoelectric point at a pH of 4.5 to 5.5, and at that point, protein hydrolysates showed lower solubility values due to weak interaction between protein molecules and water. However, increasing or decreasing the pH value from the isoelectric point improves and boosts solubility due to the high interaction between water and protein molecules [39]. According to previous studies, Noman and his colleagues reported approximately the same values of protein solubility in FPH using alcalse enzyme with sturgeon fish muscles [22]. Additionally, 85% to 95% protein solubility was observed by Kamal and others [7] using the ficin enzyme with bighead carp. Furthermore, Li and his colleagues also determined the solubility of hydrolysates using protease enzymes, and the results were fairly similar to the values in our study [40]. Finally, the difference in solubility values under different DHs might relate to the different sizes of peptide fractions formed during the hydrolysis process along with the ions of chemical groups held on the fractions [41].

#### 2.11.2. Water- and Oil-Holding Capacity

The values of OHC and WHC in fish protein hydrolysates derived from bighead carp utilizing the flavourzyme enzyme under optimum conditions of enzymatic hydrolysis are displayed in Table 1. The current results of WHC showed significant differences (*p* < 0.05) at different DHs, which were 1.97, 2.64, and 3.49 (g water/g protein) at DH 16.56%, 22.23% and 25.48%, respectively. The WHC values in this study were higher than those reported by [22] in Chinese sturgeon fish using the alcalse protease, but they were lower than the results reported by [7] in bighead carp fish using the ficin enzyme for the hydrolysis process. According to previous research, high values of WHC in hydrolysis products improve the taste of various food products supported by protein hydrolyses [6]. During enzymatic hydrolysis, different values of DH have an impact on the functionality of WHC. However, by increasing DH, there was an increase in the polar groups (NH2) and (COOH) in hydrolysates products, which influences the final values of WHC [6].

Furthermore, oil-holding capacity values were 2.85, 2.31, and 2.19 g/g protein hydrolysate at DH 16.56%, 22.23%, and 25.48%, respectively. Our results in this study were lower than those reported by Noman and others [22] regarding hydrolysates from sturgeon fish using alcalse, and they were lower than the values of OHC reported by Kamal and his colleagues in bighead carp fish hydrolysates using the ficin enzyme [7]. Relatively similar OHC results were reported by Wasswa [12], who obtained hydrolysates from grass carp skin materials. Additionally, reducing OHC values and boosting the degree of hydrolysis might contribute to the catalyzed peptides during the enzymatic hydrolysis process and its molecular mass impact, suggesting that the lower quantity of oil could be absorbed by small molecular peptides. Eventually, at DH 25.48%, OHC was 2.19 g/g protein hydrolysate, which was the lowest value. Thus, oil-holding capacity has been demonstrated to be an important functional property that has a remarkable impact on sensory and emulsification properties of food products [7].

### 2.12. Circular Dichroism Spectra

The influence of the enzymatic hydrolysis process using the flavourzyme enzyme on the protein secondary structure was investigated, and the data are shown in Table 2. CD (Selcon3) software analysis was applied to monitor and calculate the changes of α-helix, β-turn, β-sheet, and unordered structures of the hydrolysate samples. After hydrolysis using the flavourzyme enzyme, the β-turn, β-sheet, and unordered components were increased, and there was a slight increase in α-helix values in the hydrolysate samples. The current study showed that enzymatic hydrolysis changed the structural configuration of the protein, which might be caused by protein precipitation or peptide unfolding through enzymatic hydrolysis [11]. The values of the β-turn, β-sheet, and unordered components (although not α-helix) were higher than those reported by [11] using the flavourzyme protease in lupin protein isolate or soybean protein hydrolysates reported by Zhao and colleagues [42]. The α-helix value decreased from 5.3% at DH 16.56% to 4.10% at DH 22.23%, while β-sheet and β-turn values were increased with increasing DH. Furthermore, unordered values ranged from 32.1 to 34.4% in all samples. In this study, the hydrolysis process, which degraded the protein molecules into small peptides via enzymatic cleavage, led to different changes in protein structure component values, including β-sheets and β-turns, in hydrolysate products [43]. The major secondary structures (α-helix, β-sheets and β-turns) refer to regular arrangements of amino acid residues in the peptide chains consisting of hydrogens and carbonyl oxygens of the peptide backbone [44].

### 2.13. Antioxidant Activities of Fish Protein Hydrolysates

#### 2.13.1. DPPH Radical-Scavenging Activity

The radical-scavenging capacity of FPH was determined and investigated using DPPH with absorbance at 517 nm. In the DPPH assay, the violet color the DPPH solution is reduced to yellow because of the contribution of a hydrogen ion (H+), which reduces the absorbance [24]. Figure 7A displays the DPPH radical-scavenging activity values of FPH at various degrees of hydrolysis and different conversations from 1 to 20 mg/mL. The hydrolysates obtained from bighead carp using the flavourzyme enzyme revealed that DPPH radical-scavenging activity had a notable increase from 8.46%, 19.20%, and 17.08% to 66.17%, 64.95%, and 71.18% for FPH produced at DH 16.56%, 22.23%, and 25.48%, respectively, when the concentration of samples was increased from 1 mg/mL to 20 mg/mL. These values were higher than those reported by Noman [22] in hydrolysates of sturgeon fish, but they were lower than the results found by Md. Serajul [15] in Chinese grass turtles.

On the other hand, the IC_50_ value is described as the concentration of protein hydrolysates that can scavenge 50% of DPPH free radicals. The IC_50_ values of hydrolysates produced at DH 16.56%, 22.23%, and 25.48% were 7.92, 7.09, and 6.11 mg/mL, respectively. Obviously, the hydrolysates produced at DH 25.48% had the highest DPPH-scavenging activity. Furthermore, the different radical-scavenging values of hydrolysate products often related to amino acid compositions, the sequences of different peptides, and the size of peptides fractions produced under several DHs [45]. Finally, according to amino acid results shown in Table 3, of the hydrolysate samples, the different amino acid content led to various values of IC_50_ in FPH. Thus, DH 25.48% had the highest amino acid composition compared to DH 16.56% and 22.23%.

#### 2.13.2. Hydroxyl Radical-Scavenging Activity

The hydroxyl radical-scavenging assay is the most reactive oxygen and was evaluated using ascorbic acid to generate hydroxyl radicals. The data regarding hydroxyl radical-scavenging activity are displayed in Figure 7B. Based on the results, significant variation was observed among different DHs. The highest value, 69.98%, was recorded at DH 16.56%, a value of 59.07% was recorded at DH 22.23%, and a value of 53.78% was recorded at DH 25.48% at the same concentration of 15 mg/mL. According to our study, the lowest degree of hydrolysis obtained the highest hydroxyl radical-scavenging activity. Our results are similar to those reported by Md. Serajul and his colleagues in protein hydrolysates of Grass turtles [15], but they are higher than those reported by Gao [46]. On the other hand, the IC_50_ of protein hydrolysates at DH 16.56% was the best-recorded value of 10.27 mg/mL, compared with IC_50_ values at DH 22.23% and 25.48%, which were 12.86 mg/mL and 13.08 mg/mL, respectively, which did not have a clear significant difference (*p* ≤ 0.05).

#### 2.13.3. Metal Iron (Fe^+2^) Chelating Activity

Results of the investigation of iron-chelating activity of FPH produced from bighead carp using the flavourzyme enzyme at different degrees of hydrolysis are reported in Figure 7C. The current study observed that increasing hydrolysate concentration from 1 mg/mL to 20 mg/mL led to increased iron (Fe^+2^) chelating activity in the treated samples. However, all the hydrolysate samples at the highest concentration of 20 mg/mL displayed Fe^+2^-chelating activity higher than 50%. Furthermore, the chelating activity of Fe^+2^ in FPH produced at DH 16.56% and DH 22.23% demonstrated remarkable values at various concentrations of FPH in comparison with DH 25.48%. Generally, the iron metal Fe^+2^-chelating activity might be related to histidine residues along with boosting of (COOH) and (NH_2_) chemical groups in the sides of acidic and essential amino acids in the peptide components of FPH [22]. The type of amino acids considerably impacts the chelating activity of iron and copper ions. This ion-chelating property can decrease negative hydrolysate product changes [47]. The IC_50_ of FPH produced at 16.56%, 22.23%, and 25.48% were 27.61, 32.19, and 32.86 mg/mL, respectively.

## 3. Materials and Methods

### 3.1. Preparation of Raw Materials

Raw bighead carp was brought from the Yangtze River with the help of the freshwater fish institute technology in Wuxi, China. The raw fish was shipped in a specific container to the Food Processing and freshwater product laboratory at the School of Food Science and Technology. The raw fish was prepared according to Aquatic products protocol and kept at −20 °C to avoid harm until further use.

### 3.2. Enzyme and Chemicals

Flavourzyme protease enzyme with approximate optimum conditions (activity 30 U/mg, pH 6.0, temperature reaction 50 °C) was purchased from (Wuxi Decheng Lebang Biotechnology Co., Ltd.) located at (99 Jinxi Road, Binhu District, Wuxi, China), and the flavourzyme was maintained at 4 °C until use. Solutions and reagents, including alkaloids and acids, were provided by Sinopharm Chemical Company (Wuxi, China). The ferrozine, DPPH, and ABTS reagents were purchased from Sigma-Aldrich (Shanghai, China). Other chemical components were analytical grade.

### 3.3. The Production Process of Protein Hydrolysates

FPH from bighead carp was generated using the flavourzyme enzyme following the process observed by Noman and others [8] with minor modifications. Single-factor experiments (pH, temperature, E/S ratio, and reaction time) were applied to achieve the optimum parameters, as demonstrated in Table 4. The raw fish was minced and mixed with 50 mM sodium phosphate buffer to keep a constant pH during the hydrolysis process. The hydrolysis reaction was accomplished in a jacketed vessel (500 mL) that was stirred during the hydrolysis time, and to control the temperature, a circulated water bath (Blue pard Technical. Co., Shanghai, China) was connected with the 500 mL jacketed vessels. To deactivate flavourzyme enzymatic activity, hydrolysis samples were heated to 85 °C for 30 min in the water bath. The hydrolysis was left at room temperature and then centrifuged for 20 min at a speed of 8000 rpm at 4 °C. Eventually, the upper layer (supernatant) was collected and freeze-dried at −55 °C in a vacuum, and the hydrolysate product was then kept at −18 °C for subsequent experiments.

### 3.4. Determination of Degree of Hydrolysis

The DH was reported as the percent of the free amino group derived from hydrolysates samples, which was measured using a titration procedure as conducted in [8], with minor adjustments. A determined quantity of mixed hydrolysates (1.4 to 1.8 g) was mixed with 47 to 50 mL of distilled water. The hydrolysate sample was then adjusted to pH 7.0 by adding 0.1 N NaOH, and 10 mL of formaldehyde 37% to 40% (*v*/*v*) was then added and maintained for 10 to 12 min at room temperature. Using 0.1 N of NaOH, the titration reached an endpoint at pH 8.5. The free amino group was then calculated using the consumed volume of NaOH. The Kjeldahl technique determined the total nitrogen (TN) [48]. The following equations were used to calculate the DH and the free amino groups:Free amino groups (%) = (V × C × 14.007/M × 1000) × 100(1)
DH = (%free amino groups/%total nitrogen) × 100(2)
where V = volume consumed of NaOH (mL); C = NaOH concentration (0.1 M); and M = the specific amount of hydrolysate in the treated sample (g).

### 3.5. Yield

The yield of FPH was measured and calculated according to Romadhoni [49] and is given by the following equation:Yield (%) = [weight of hydrolysate product (g)/weight of raw materials (g)] × 100(3)

### 3.6. Chemical Composition Analysis

FPH was produced under enzymatic hydrolysis, and raw materials were individually analyzed. Official procedures of AOAC 985.2912, 960.39, 928.08, and 923.03 were applied to evaluate the contents of moisture, fats, protein, and ash, respectively [50]. Briefly, 2 to 3 g was weighed in the crucible and heated in a muffle furnace at 450 to 550 °C for 6 to 7 h until steady weight and reddish ash were noticed. For lipid extraction, a sample (3 g) was dissolved and extracted in petroleum ether for 2 to 3 h in determiner equipment (Auto Lipid determiner, Xian Jian Instruments Co., Shanghai, China). The crude protein was evaluated using the Kjeldahl determiner (DK-3400/FOSS, Hilleroed, Denmark) and multiplying the total nitrogen obtained by 6.25. To determine the moisture content, the evaporating process was used for 2.5 to 3 h at 100 to 105 °C until it reached a steady weight.

### 3.7. Water Activity

Water activity was assessed using a lab instrument (Novasina AG, 100–240V, Switzerland) at 25 °C with an accuracy of 0.001. FPH powder was placed in a specimen chamber and maintained until equilibrium. The values were carried out in triplicate.

### 3.8. Color Measurement

A digital lab colorimeter (UltraScan PRO, D65, Reston, VA, USA) was used to detect the color of the hydrolysate product. The targeted hydrolysates were placed at the port of the hunter colorimeter. The values of color detection L*, a*, and b* were investigated; L* represents lightness; a* represents redness to green; b* represents yellowness to blue, while ΔE was determined as the total color variation.

### 3.9. Amino Acid Determination

A total of 100 to 110 mg of hydrolysates were digested using 6 M hydrochloride-HCl at 110 to 120 °C in an air oven for 20 to 24 h. After that, the samples were cooled under a room atmosphere, and 4.8 mL of 10 M NaOH was then added. The targeted samples were raised up to 25 mL using purified water. Using the Whatman filter, the samples were filtered and centrifuged for 12 min at 10,000 rpm. Amino acids were detected using HPLC (model 1100, Agilent Technologies, Santa Clara, CA, USA); the RP mode was 180 mm × 4.6, Agilent, Zorbax 80 A, and C18 column at 37 to 40 °C, and the detection value was 338 nm under a steady flow rate 1 mL/min. The mobile phase (A) was 7.35 mM/L of sodium acetate/triethylamine/tetrahydrofuran (500:0.12:2.5, *v*/*v*/*v*), and acetic acid was added to modify the pH to 7.2. The mobile phase (B) was 7.35 mM/L of sodium acetate/methanol/acetonitrile (1:2:2, *v*/*v*/*v*) with a pH of 7.2. Amino acid values were observed as a gram of amino acid per 100 g of protein.

### 3.10. Molecular Mass Distribution Analysis

The analysis of the molecular mass distribution of the protein hydrolysate obtained by enzyme hydrolysis at 1, 3 and 6 h compared to the raw material was investigated according to the guidance of Liu, Lin, Lu, and Cai [51], with slight amendments. Waters HPLC model (Waters-1525, Milford, MA, USA) system with TSKgel 2000 SWXL (300 × 7.8 mm) column manufactured (Tosoh, Tokyo, Japan) with mobile phase acetonitrile/water/trifluoroacetic acid (TCA) 45/55/0.1 (*v*/*v*) was utilized. The samples were dissolved in mobile phase buffer before injection and were then centrifuged for 15 min at 9000 rpm, after which they were filtered using 0.22 µm filters. Finally, the targeted samples were monitored at 220 nm, the column temperature was 30 °C, and the flow rate was at 0.5 mL/min. The standards for the calibration curve of the molecular mass profile were Cytochrome C (12,384 Da), bacitracin (1422 Da), Gly-Gly-Try-Arg (451 Da), and Gly-Gly-Gly (189 Da).

### 3.11. Electrophoretic Profile Using SDS-Page

Sodium dodecyl sulfate-polyacrylamide gel electrophoresis (SDS-PAGE) was used to detect the molecular mass of the hydrolysate samples based on the procedure described by Laemmli [52]. The stacking and separating gels were 10% and 18% acrylamide, respectively. The hydrolysate samples (5 to 7 mg/mL) were mixed in a buffer with a pH of 7.2 to 7.4, and then the mixtures (20 to 25 µL) were loaded into the gel. The hydrolysates in the gel were mobilized using electrophoresis (PowerPac^TM^, Hercules, CA, USA). Ultimately, the gel was stained and then de-stained, and a protein marker (1.2–45 kDa) was used to determine the molecular weight.

### 3.12. Scanning Electron Microscopy

A scanning electron microscope (Hitachi-High-Tech’s-SU1510-Minato-Ku, Tokyo, Japan) instrument was used to observe the morphological properties of the hydrolysate products. Before loading into the SEM machine, the samples were coated, and the images were then witnessed at volatge 1.0 KV by a secondary electron image. A 10.20 mm Ricoh Camera with 600× magnification was applied for image scanning.

### 3.13. Fourier Transform Infrared Spectroscopy

The difference in the hydrolysates’ chemical bonds was detected using FTIR (PerkinElmer Technology, Waltham, MA, USA). Briefly, the FTIR spectra of the samples obtained under different degrees of hydrolysis and a total of 32 scans were implemented at a spectral resolution of 4 cm^−1^, with a wave number range of 400–4000 cm^−1^ for spectrum. The transform infrared spectroscopy was applied to evaluate the impact of enzymatic hydrolysis on the secondary protein structure of hydrolysates. Different absorption peaks were studied and analyzed. To carry out the FTIR spectrum, a tiny amount of hydrolysate products was smoothly placed on the targeted crystal, and the pressure tool was then pushed to lightly fix the products on the diamond crystal. FTIR records were in triplicate, and the average of each sample was calculated.

### 3.14. Functional Characteristics of FPH

#### 3.14.1. Solubility

The solubility of FPH produced under enzymatic hydrolysis was evaluated based on the procedure recorded by Jain and Anal [53], with minor amendments. A mass of 150 to 200 mg of hydrolysate was mixed with 20 mL of deionized water, and then, by utilizing 0.1 M NaOH or 0.1 M HCl, the pH of the solution was modified from 2 to 10. The samples were incubated at 28 °C with constant stirring at 150 rpm for 25 min, and it was then centrifuged at 8500× *g* for 25 min. The Kjeldahl determiner determined the content of protein in the supernatant and hydrolysate samples, and the following equation gave the solubility:Solubility% = (protein content in the supernatant/protein content in FPH) × 100(4)

#### 3.14.2. Water-Holding Capacity

The determination of hydrolysates’ water-holding capacity (WHC) obtained under different DHs was described and guided by [12] with some modifications. A total of 10 mL of Distilled water was mixed with 0.5 g hydrolysate and then dispersed using a vortex for 55 s. The dispersion was kept for 5 h at 25 °C, and it was then centrifuged at 4500 rpm for 25 min. Next, the supernatant was filtered, and the volume recovered was measured carefully. Finally, the difference between the distilled water volume and the volume of the supernatant was calculated. The values of WHC were presented as mL of H_2_O absorbed per gram of FPH product.

#### 3.14.3. Oil-Holding Capacity

The oil-holding capacity of various hydrolysate products was determined based on the method reported by [12] with some amendments. A volume of 10 mL of sunflower oil was mixed with hydrolysates (0.5 g) and vortexed for 40 s in the tube. After that, the samples were centrifuged at 4500 rpm for 25 min and measured. The absorption of hydrolysate samples was measured based on weight variance, and the OHC value was calculated as grams of oil absorbed per gram of hydrolysate powder.

### 3.15. Protein Secondary Structure

Circular dichroism (CD) spectroscopy was carried out using Chirascan; Applied Photophysics Ltd., Surry, UK to evaluate the secondary structural changes of FPH under various degrees of hydrolysis with four accumulations in the far UV area (190–260 nm). Briefly, the sample was prepared in phosphate buffer with a pH of 6.9 and a concentration of 0.3–0.5 mg/mL and was then placed in a cell with 0.1 mm optical path length. The hydrolysate products’ secondary structural characteristics (α-helix, β-strand, β-turns, and unordered coil) were recorded, and the values were revealed in parentage using (Selcon3) software. All values were recorded in triplicate.

### 3.16. Antioxidant Properties of FPH

#### 3.16.1. DPPH Radical-Scavenging Activity

The radical-scavenging activity of FPH was measured according to the protocol reported by [54] with modifications. In short, FPH powder was diluted in various concentrations from 1 to 20 mg/mL. Next, 100 μL of the hydrolysate samples was added to 1.70 mL of methanolic DPPH (60 μmol) in a microplate well. The mixture was incubated for 25 min at room temperature (25 °C) in the dark; the control was DPPH solution mixed with 100 μL of distilled water. A wavelength of 517 nm was used to observe the absorbance. The absorbance results were converted to DPPH scavenging activity (%) using the following equation:(5)$$\mathrm{DPPH}\;\mathrm{scavenging}\;\mathrm{activity}\%\;=\left[\left(A_{0}-\;A₁\right)/A_{0}\right]\times 100$$
where A0 is the control absorbance and A1 is the sample absorbance.

IC50 values were calculated based on the concentration of FPH against the inhibition at (50%) implemented in Excel (2013).

#### 3.16.2. (Fe^2+^) Chelating Activity

The iron-chelating effect of bighead carp protein hydrolysates was measured according to [23] with slight changes. One milliliter of the samples at various concentrations (1, 5, 10, 15, and 20 mg/mL) was mixed with 0.2 mL of ferrozine (5 mM) and 0.1 mL of FeCl2 (2 mM). The mixed solution was maintained at 25 °C for 15 min, and then the absorbance values were determined at 562 nm. The blank was prepared using purified water instead of hydrolysate solution, whereas the control was papered without the addition of ferrozine. The chelating activity of (Fe^2+^) was calculated as follows:Chelating activity (Fe^2+^)% = [1 − (A_sample_ − A_sample-control_)/A_blank_] × 100(6)

#### 3.16.3. Hydroxyl Radical Scavenging Assay

The hydroxyl radical scavenging activity of FPH was measured following the procedure described by [55] with some changes. In 24 microplates, 100 μL of hydrolysate samples at different concentrations (1–20 mg/mL) was mixed with 200 μL of FeSO_4_ (2 mM), 200 μL of 9 mM salicylic acid in 96% ethanol, and 200 μL of H_2_O_2_ (0.03% *v*/*v*) and was then incubated at 37 °C for 75 min. The absorbance was achieved at 536 nm against a blank sample (distilled water instead of FPH solutions).

### 3.17. Statistical Analysis

All samples were assessed in triplicate (n = 3), and the data were observed as mean value ± standard deviation (±SD). One-way analysis of variance (ANOVA) was implemented to evaluate significant differences. Duncan’s multiple range test was performed to analyze various ranges between means. Statistical analyses were conducted utilizing SPSS ver.19 (SPSS. Chicago, IL, USA), and *p* ≤ 0.05 was considered statistically significant.

## 4. Conclusions

Fish protein hydrolysates were obtained from bighead carp *(Hypophthalmichthys nobilis)* using the flavourzyme enzyme under an optimal temperature of 50 ℃, pH of 6.5, and 4% E/S ratio. The current findings demonstrated that functional, structural, and antioxidant characteristics of FPH were remarkably influenced by various degrees of hydrolysis. Regarding the potential application in the food industry, the functional and structural properties (water- and oil-holding capacity, solubility, and secondary structure) of fish protein hydrolysates showed a significant enhancement along with good antioxidant activity (i.e., hydroxyl radical-scavenging and (1,1-diphenyl-2-picrylhydrazyl) DPPH-scavenging activities) and Fe^+2^-chelating activity. This work analyzed the protein secondary structure and FTIR spectrum of hydrolysate products. Furthermore, the hydrolysate products showed excellent nutritional properties and promising antioxidant activities, suggesting their potential use as a protein additive in various food applications. Further studies regarding peptide fractions and applications in medical and pharmaceutical areas will be conducted.

## Figures and Tables

**Figure 1 molecules-28-00519-f001:**
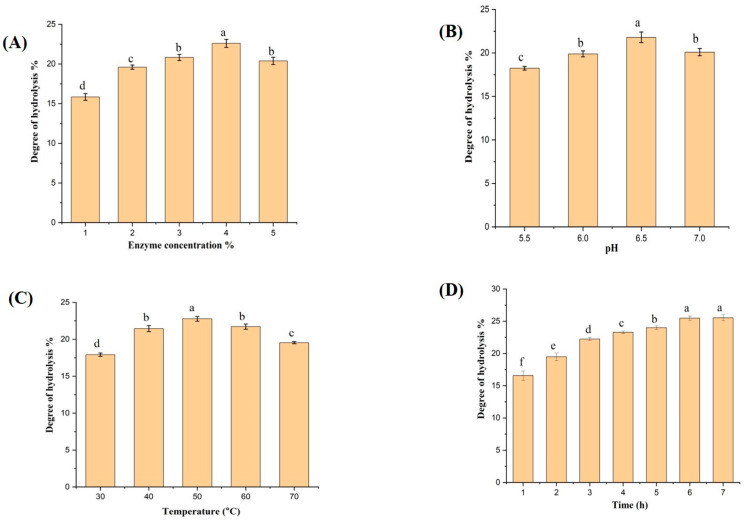
Influence of enzymatic hydrolysis conditions (**A**) E/S ratio; (**B**) pH; (**C**) temperature; and (**D**) time of hydrolysis reaction on degree of hydrolysis (DH). The values represent mean ± SD (n = 3), and the various letters indicate the values are significantly different (*p* ≤ 0.05).

**Figure 2 molecules-28-00519-f002:**
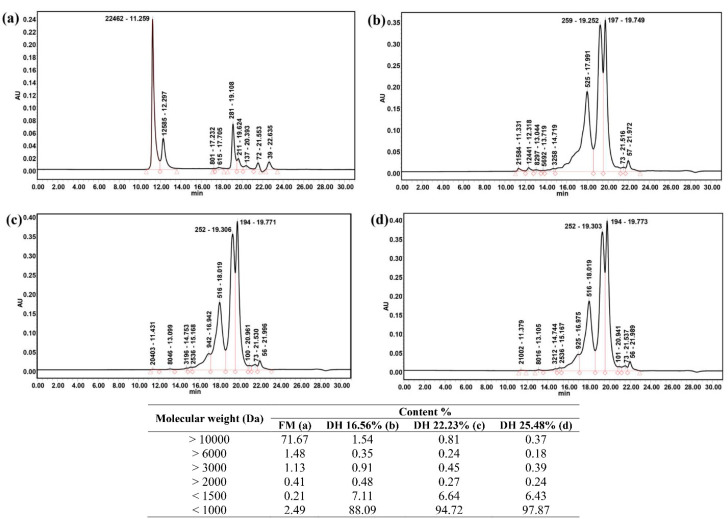
Molecular weight distribution of the fresh and hydrolysate products derived from fish bighead carp using flavourzyme protease under three different degrees of hydrolysis: (**a**) fresh sample; (**b**) DH 16.56% (1 h); (**c**) DH 22.23% (3 h); and (**d**) DH 25.48% (6 h).

**Figure 3 molecules-28-00519-f003:**
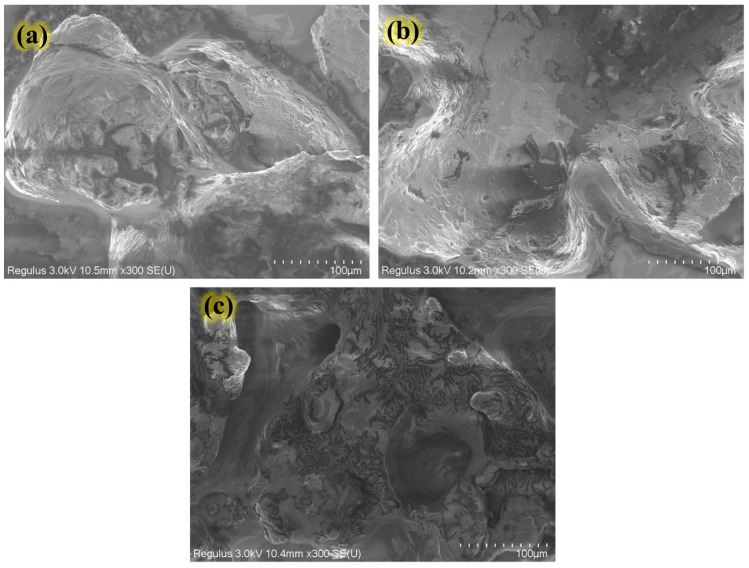
Scanning electron microscopy (SEM) of protein hydrolysates at various degrees of hydrolysis; (**a**) DH 16.56%, (**b**) DH 22.23%, and (**c**) DH 25.48%.

**Figure 4 molecules-28-00519-f004:**
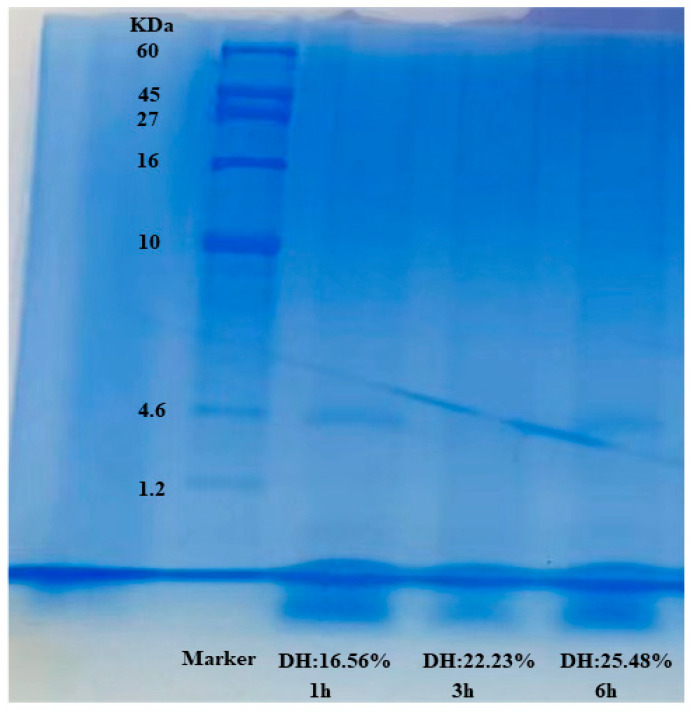
SDS-PAGE pattern of protein hydrolysates under different degrees of hydrolysis.

**Figure 5 molecules-28-00519-f005:**
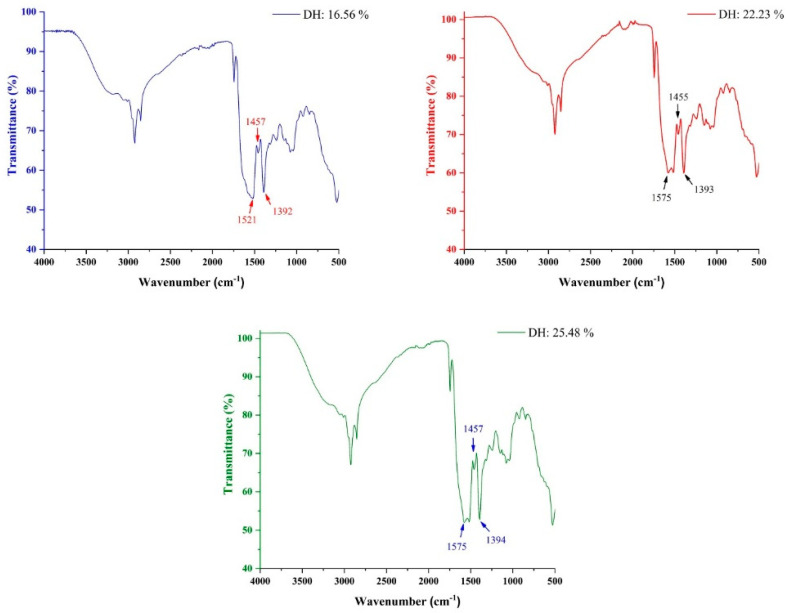
FTIR spectrum of FPH obtained at different degrees of hydrolysis using the flavourzyme enzyme. The various values on the graphs indicate the changes in amide bands I, II, and III.

**Figure 6 molecules-28-00519-f006:**
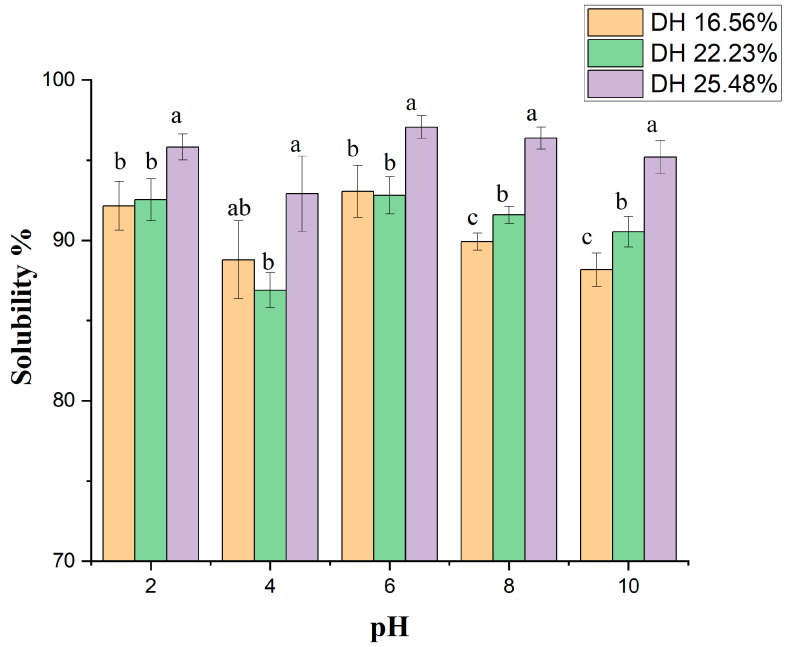
Protein solubility of hydrolysates obtained at different degrees of hydrolysis at pH values ranging from 2 to 10. The values represent mean ± SD (n = 3), and various letters indicate that the values are significantly different (*p* ≤ 0.05).

**Figure 7 molecules-28-00519-f007:**
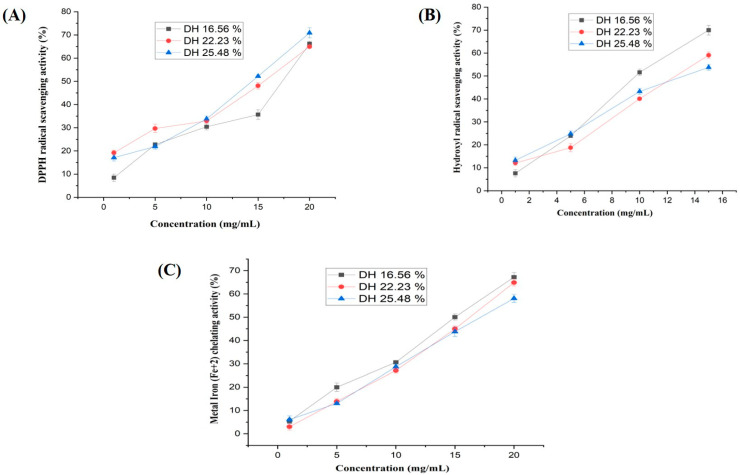
Antioxidant assays of FPH produced under optimal parameters at different degrees of hydrolysis, (**A**) DPPH radical-scavenging activity, (**B**) hydroxyl radical-scavenging assay, and (**C**) Fe^+2^-chelating activity. The results are displayed as mean ± SD with triplicates.

**Table 1 molecules-28-00519-t001:** Chemical composition, color measurement, yield, and water activity of protein hydrolysates (FPH) prepared using the flavourzyme enzyme under optimum conditions (n = 3, mean ± SD).

		Degree of Hydrolysis	
Parameters	16.56%	22.23%	25.48%
Hydrolysate yield	13.08 ± 0.58 ^c^	17.10 ± 0.36 ^b^	17.83 ± 0.67 ^a^
Moisture	3.72 ± 0.09 ^a^	3.64 ± 0.05 ^a^	3.77 ± 0.09 ^a^
Protein	83.61 ± 1.25 ^c^	85.94 ± 0.77 ^b^	88.19 ± 2.02 ^a^
Fat	2.98 ± 0.09 ^a^	3.34 ± 0.12 ^a^	2.04 ± 0.05 ^b^
Ash	6.21 ± 0.12 ^a^	5.92 ± 0.10 ^a^	4.85 ± 0.23 ^b^
Color measurement			
L*	86.89 ± 0.52 ^a^	84.06 ± 0.24 ^b^	83.98 ± 0.16 ^b^
a*	1.39 ± 0.08 ^b^	1.43 ± 0.05 ^b^	1.92 ± 0.11 ^a^
b*	15.81 ± 0.37 ^b^	17.42 ± 0.18 ^a^	17.97 ± 0.21 ^a^
∆E	18.11 ± 0.09 ^a^	17.28 ± 0.05 ^b^	18.51 ± 0.13 ^a^
Water activity	0.36 ± 0.00 ^a^	0.34 ± 0.00 ^a^	0.27 ± 0.01 ^b^
WHC (g/g FPH)	1.97 ± 0.09 ^c^	2.64 ± 0.17 ^b^	3.49 ± 0.25 ^a^
OHC (g/g FPH)	2.85 ± 0.07 ^a^	2.31 ± 0.15 ^b^	2.19 ± 0.11 ^b^

Different letters indicate significantly different values (*p* ≤ 0.05). All values represent mean of triplicate measurements, mean ± SD (n = 3). L*, a*, b*, ΔE values are standards of color measurements. L* refers to lightness, ranging from 0 to 100, a* refers to (green to red), and b* refers to (blue to yellow). WHC indicates water-holding capacity (g water/g hydrolysate). OHC indicates oil-holding capacity (g oil/g hydrolysate).

**Table 2 molecules-28-00519-t002:** Secondary structure composition of bighead carp protein hydrolysates obtained using the flavourzyme enzyme at different degrees of hydrolysis.

CD Values		Hydrolysates	
	16.56%	22.23%	25.48%
α-Helix (%)	5.3 ± 0.18 ^a^	4.1 ± 0.11 ^b^	4.15 ± 0.25 ^b^
β-Sheets (%)	40.1 ± 0.64 ^b^	42.5 ± 1.82 ^a^	42.4 ± 1.12 ^a^
β-Turn (%)	20.2 ± 0.48 ^b^	21.4 ± 1.09 ^a^	21.3 ± 0.51 ^a^
Unordered (%)	34.4 ± 1.30 ^a^	32.1 ± 1.44 ^b^	32.1 ± 0.60 ^b^

Various letters in the same row refer to significant differences (*p* < 0.05) between hydrolysate samples under different degrees of hydrolysis.

**Table 3 molecules-28-00519-t003:** Amino acid composition of bighead carp protein hydrolysates (g/100 g protein) obtained using the flavourzyme enzyme under optimum hydrolysis conditions.

Amino Acids		Hydrolysates	
**Essential amino acids (EAAs)**	DH 16.56% (1 h)	DH 22.23% (3 h)	DH 25.48% (6 h)
Histidine	2.87 ± 0.08 ^b^	2.89 ± 0.07 ^b^	3.12 ± 0.02 ^a^
Threonine	4.10 ± 0.04 ^b^	4.90 ± 0.04 ^a^	4.88 ± 0.02 ^a^
Valine	4.67 ± 0.06 ^b^	4.68 ± 0.03 ^b^	5.13 ± 0.05 ^a^
Methionine	2.83 ± 0.04 ^b^	2.90 ± 0.06 ^b^	3.11 ± 0.08 ^a^
Phenylalanine	3.63 ± 0.04 ^c^	3.78 ± 0.02 ^b^	4.12 ± 0.03 ^a^
Isoleucine	4.46 ± 0.02 ^b^	4.51 ± 0.01 ^b^	4.90 ± 0.09 ^a^
Leucine	6.59 ± 0.02 ^b^	6.66 ± 0.04 ^b^	6.84 ± 0.05 ^a^
Lysine	7.11 ± 0.03 ^b^	7.20 ± 0.08 ^b^	7.88 ± 0.04 ^a^
Arginine	6.38 ± 0.02 ^b^	6.41 ± 0.01 ^b^	6.91 ± 0.02 ^a^
Tyrosine	2.67 ± 0.03 ^b^	2.92 ± 0.05 ^a^	2.94 ± 0.09 ^a^
**Non-essential amino acids (NAAs)**			
Cysteine	0.21 ± 0.03 ^a^	0.22 ± 0.02 ^a^	0.26 ± 0.02 ^a^
Aspartic acid	9.11 ± 0.02 ^b^	9.20 ± 0.07 ^b^	9.44 ± 0.09 ^a^
Glutamic acid	12.47 ± 0.04 ^a^	12.45 ± 0.02 ^a^	12.29 ± 0.07 ^b^
Serine	2.37 ± 0.03 ^a^	2.38 ± 0.08 ^a^	2.32 ± 0.05 ^a^
Glycine	3.93 ± 0.05 ^b^	4.20 ± 0.01 ^a^	3.89 ± 0.04 ^b^
Proline	2.41 ± 0.03 ^b^	2.67 ± 0.06 ^a^	2.61 ± 0.07 ^a^
Alanine	5.30 ± 0.02 ^b^	5.34 ± 0.01 ^a^	5.26 ± 0.02 ^b^
TEAA	45.34 ± 0.58 ^c^	46.93 ± 0.28 ^b^	49.81 ± 0.19 ^a^
TNAA	35.81 ± 0.16 ^b^	36.11 ± 0.21 ^ab^	36.22 ± 0.29 ^a^
TAA	81.15 ± 0.41 ^c^	83.05 ± 0.54 ^b^	86.03 ± 0.34 ^a^
TEAA/TAA%	55.87 ± 0.33	56.50 ± 0.45	57.91 ± 0.52

Different letters indicate that the values are significantly different (*p* ≤ 0.05). Values represent the mean of triplicate measurements, mean ± SD (n = 3). TAA indicates total amino acids, TEAA indicates total essential amino acids, TNAA indicates total non-essential amino acids, and TEAA/TAA% indicates the percent ratio of total essential amino acids to total amino acids.

**Table 4 molecules-28-00519-t004:** The parameters applied to obtain optimal enzymatic hydrolysis conditions to produce protein hydrolysates from bighead carp *(Hypophthalmichthys nobilis)* using the flavourzyme enzyme.

Parameters	Levels
pH	5.5, 6, 6.5, and 7
Temperature (°C)	30, 40, 50, 60, and 70
E/S ratio (%)	1, 2, 3, 4, and 5
Hydrolysis time (h)	1, 2, 3, 4, 5, 6, and 7

## Data Availability

Not applicable.

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
