# Peer review of "Influence of the Enzymatic Hydrolysis Using Flavourzyme Enzyme on Functional, Secondary Structure, and Antioxidant Characteristics of Protein Hydrolysates Produced from Bighead Carp (Hypophthalmichthys nobilis)"

_molecules, 2023, doi:10.3390/molecules28020519_

Round 1

Reviewer 1 Report

The work was presented very well with obvious methods and the data are clear and easy to understand with sufficient discussion. But have some points need to be clarified:

1-      Line 68 to 71, the enzymatic hydrolysis process targeting seafood protein had the remarkable potential to apply in different food industries compared with other hydrolysis methods. The enzymatic hydrolysis might provide prospective nutritional and functional products for wide uses in food applications. (Please provide a reference).

2-      Line 200 to 202, The mobile phase (A) and The mobile phase (B), How to adjust 2 mobile phases to obtain total amino acids

3-      Line 491 to 495, could you explain more about how different amide bonds give different absorption values under the FTIR machine

4-      Check the references (5, 9, 39, and 41) and modify them according to the Journal style.   

5-      Modify your conclusion and add more details about the body research. 

6-      The English language needs to improve. This appears to be a minor revision needed for this manuscript.

Author Response

Author’s Reply to the Review Report (Reviewer 1)

  • Line 68 to 71, the enzymatic hydrolysis process targeting seafood protein had the remarkable potential to apply in different food industries compared with other hydrolysis methods. The enzymatic hydrolysis might provide prospective nutritional and functional products for wide uses in food applications. (Please provide a reference).

Response: thanks for your valuable comment, already added the reference line 62.

  • Line 200 to 202, The mobile phase (A) and The mobile phase (B), How to adjust 2 mobile phases to obtain total amino acids

Response: thanks for your valued question, the adjusting mobile phase 1 and mobile phase 2 occurs at the HPLC instrument during the process of amino acids determination.

  • Line 491 to 495, could you explain more about how different amide bonds give different absorption values under the FTIR machine

      Response: thanks for your suggestion, actually during the enzymatic hydrolysis under   different    times, various amide bonds formed and then under the FTIR absorption peaks can give different values according to the hydrolysis time, the type substrate, the catalysis site of the protease with the substrate and the type of protease used in the hydrolysis process.

  • Check the references (5, 9, 39, and 41) and modify them according to the Journal style.

      Response: thanks for your comment, we modified according to your valuable suggestion.

  • Modify your conclusion and add more details about the body research.

     Response: thanks for your valued suggestion, we modified the conclusion part according to your advice.

  • The English language needs to improve. This appears to be a minor revision needed for this manuscript.

   Response: thanks for your suggestion, we appreciate your effort in this article, we modified some English errors in the manuscript.

Reviewer 2 Report

1. The authors need to improve the language level further.

2. The temperature, pH and enzyme/material ratio were fixed through a single-factor experiment, and then the time was controlled to obtain enzymolysis liquid at different times. Why choose enzymatic hydrolysates at different times? Is it because the degree of hydrolysis is different at different times? What is the difference between the sample with a similar degree of hydrolysis obtained under other conditions?

3. Why Flavor Protease?

4. Please keep the words in the full text consistent. For example, color or colour.

5. Please note the correct reference format, such as line 198.

6. Does the hydrolysate with a high degree of hydrolysis contain bitter peptides and other undesirable components?

Author Response

Author’s Reply to the Review Report (Reviewer 2)

  • The authors need to improve the language level further.

Response: thanks for your valued suggestion, we tried to improve the language of the manuscript using some language software.

  • The temperature, pH and enzyme/material ratio were fixed through a single-factor experiment, and then the time was controlled to obtain enzymolysis liquid at different times. Why choose enzymatic hydrolysates at different times? Is it because the degree of hydrolysis is different at different times? What is the difference between the samples with a similar degree of hydrolysis obtained under other conditions?

Response: thanks for your comment, choosing different times was because different degrees of hydrolysis during the enzymatic process, and in this research we focused about the functional properties with antioxidants activities in the hydrolysates products. (About difference between the samples with a similar degree of hydrolysis obtained under other conditions), we didn’t analyze the samples you mentioned yet but it’s going to be r the further investigation of next research.

  • Why Flavor Protease?

Response: thanks for your comment, flavourzyme enzyme is widely and diversely used for protein hydrolysis in industrial and research applications, also flavourzyme showed a remarkable degree of hydrolysis values and the hydrolysates products obtained using flavourzyme had a significant potential application in different food industries.

  • Please keep the words in the full text consistent. For example, color or colour.

Response: thanks for your suggestion, we modified according to your comment.

  • Please note the correct reference format, such as line 198.

Response: thanks for your suggestion, we modified the references according to the journal style.

  • Does the hydrolysate with a high degree of hydrolysis contain bitter peptides and other undesirable components?

Response: thanks for your inquiry, we didn’t analyze the bitterness of hydrolysate products in this work but will take it into consideration for further investigation and experiments, highly appreciate your question.

Round 2

Reviewer 2 Report

It can be accepted in the present form.